# Partial Laryngectomy for pT4a Laryngeal Cancer: Outcomes and Limits in Selected Cases

**DOI:** 10.3390/cancers15102861

**Published:** 2023-05-22

**Authors:** Giovanni Succo, Andy Bertolin, Izabela Costa Santos, Martina Tascone, Marco Lionello, Marco Fantini, Andressa Silva de Freitas, Ilaria Bertotto, Andrea Elio Sprio, Giuseppe Sanguineti, Fernando Luiz Dias, Giuseppe Rizzotto, Erika Crosetti

**Affiliations:** 1Otorhinolaryngology Unit, San Giovanni Bosco Hospital, 10154 Turin, Italy; giovannisucco@hotmail.com (G.S.); marcofantini8811@hotmail.it (M.F.); 2Oncology Department, University of Turin, 10124 Torino, Italy; 3Otorhinolaryngology Unit, Vittorio Veneto Hospital, AULSS2 Treviso, 31029 Vittorio Veneto, Italy; andy.bertolin@aulss2.veneto.it (A.B.);; 4Brazilian National Cancer Institute, Rio de Janeiro 20230-130, RJ, Brazil; 5Radiology Service, Candiolo Cancer Institute FPO IRCCS, Candiolo, 10060 Turin, Italy; 6Department of Research, ASOMI College of Sciences, 2080 Marsa, Malta; a.sprio@asomi-osteopatia.com; 7Department of Radiation Oncology, IRCCS Regina Elena National Cancer Institute, 00144 Rome, Italy

**Keywords:** laryngeal cancer, partial laryngectomy, OPHL, T4 laryngeal cancer, laryngeal preservation, radiotherapy

## Abstract

**Simple Summary:**

The results of this large series highlight the good onco-functional results of selected pT4a laryngeal tumors treated with open partial horizontal laryngectomies (OPHL). The best cases to be treated with OPHLs are the low-volume pT4a laryngeal tumors, with minimal or absent cartilage destruction. The level of standardization of these procedures’ indications should allow consideration of open partial horizontal laryngectomy as a valid therapeutic option in case of a patient’s absolute refusal of total laryngectomy or non-surgical protocols with concomitant chemo-radiotherapy. Based also on the obtained results in the treatment of selected pT4a, such a surgical strategy should no longer be considered anecdotal. Extending the limit of resection and including a large part of the cricoid cartilage and one crico-arytenoid unit (type III OPHL + CAU) expanded the indications of type II OPHL + ARY, allowing for a safer resection of advanced and challenging laryngeal tumors.

**Abstract:**

A large multi-institutional case series of laryngeal cancer (LC) T4a was carried out, including 134 cases treated with open partial horizontal laryngectomies (OPHL) +/− post-operative radiation therapy (PORT). The goal was to understand better whether OPHL can be included among the viable options in selected pT4a LC patients who refuse a standard approach, represented by total laryngectomy (TL) + PORT. All 134 patients underwent OPHL type I (supraglottic), II (supracricoid), or III (supratracheal), according to the European Laryngological Society Classification. Comparing clinical and pathological stages showed pT up-staging in 105 cases (78.4%) and pN up-staging in 19 patients (11.4%). Five-year data on overall survival, disease-specific survival, disease-free survival, freedom from laryngectomy, and laryngo-esophageal dysfunction-free survival (rate of patients surviving without a local recurrence or requiring total laryngectomy and without a feeding tube or a tracheostomy) were, respectively, 82.1%, 89.8%, 75.7%, 89.7%, and 78.3%. Overall, complications were observed in 22 cases (16.4%). Sequelae were observed in 28 patients (20.9%). No patients died during the postoperative period. This large series highlights the good onco-functional results of low-volume pT4a laryngeal tumors, with minimal or absent cartilage destruction, treated with OPHLs. The level of standardization of the indication for OPHL should allow consideration of OPHL as a valid therapeutic option in cases where the patient refuses total laryngectomy or non-surgical protocols with concomitant chemo-radiotherapy.

## 1. Introduction

Laryngeal cancer (LC) is one of the most common head and neck cancers, accounting for 2% of all malignant neoplasms and approximately 28% of those in the head and neck region, with 110,000–130,000 new cases expected annually worldwide [1].

In the advanced stage, T- and N-status negatively affect overall survival (OS), recognized as independent prognostic factors [2,3].

Current guidelines, derived from large retrospective series, have shown that total laryngectomy (TL) + postoperative radiotherapy (PORT) is the gold standard treatment for cT4a tumors [4,5,6]. Despite this, in many cases and facing a patient’s refusal of surgery, organ preservation protocols are often offered to treat locally advanced LC, leaving TL as a further salvage option in case of treatment failure [7].

Meanwhile, the potentiality of open partial horizontal laryngectomy (OPHL) for managing T3/T4 LCs has been highlighted following rigorous patient selection.

In 2014, the European Laryngological Society proposed a classification of the more commonly adopted horizontal partial laryngectomies according to the extent of resection, including three types of OPHL: Type I—supraglottic, Type II—supracricoid, and Type III—supratracheal [8]. The latter, described in 2006, is based on resection of the entire glottic and subglottic sites and the thyroid cartilage, sparing both or at least one functioning cricoarytenoid unit [9]. The indications of OPHL type II are essentially T2/T3 glottic/transglottic cancers without or with minimal subglottic extension and without fixed arytenoids [10,11]. The glottic/transglottic T3 category with subglottic extension +/− fixed arytenoid and T4a with limited anterior or lateral extralaryngeal extension represent the actual core indication for OPHL Type III [12]. Supratracheal partial laryngectomies (STPLs) allow sparing laryngeal function without compromising loco-regional control during long-term follow-up [13,14]. Extending the inferior limit of resection to include a large part of the cricoid cartilage and one crico-arytenoid unit, STPLs expanded the indications compared to supracricoid partial laryngectomies (SCPLs) [15].

OPHLs emerged as viable surgical options even for advanced tumors, allowing for excellent locoregional control and maintenance of a functional larynx. The main clinical difficulty in selecting advanced tumors lies in the preoperative workup. The sensitivity and specificity of imaging techniques (computed tomography (CT) and magnetic resonance imaging (MRI)) in detecting cartilage infiltration and minimal extra laryngeal extension through the membranes do not reach complete diagnostic accuracy [16].

Factors that should be considered before planning an OPHL in a patient with locally advanced LC are represented by favorable tumor subgroups based on the absence of extra laryngeal extent, absence of N, patient’s age/performance/functional status, absence of comorbidities, compliance with a sometimes-challenging rehabilitation protocol, and the plausible non-need for adjuvant radiotherapy (RT).

Although OPHL has successfully treated many cases of T4a reported in mono institutional series, current evidence-based guidelines do not mention this surgical option for managing advanced LC cases, even when the patient refuses radical surgery.

Recently, Succo et al. systematically analyzed the oncological results obtained in different subcategories of cT3–cT4a LCs treated by OPHLs using the principle of modular extension surgery. The authors found that anterior pT4a tumors with full-thickness involvement of the thyroid lamina or with spreading through the cricothyroid membrane characterized by minimal extra laryngeal extension were the most likely to be treated by OPHL, showing a good probability of success [15].

In clinical practice, such cases are similar to ones that current guidelines consider amenable to a nonsurgical organ-sparing protocol if the patient refuses TL. Offering the OPHL option in selected cases can be considered not only because of the good prognosis but also for the good functional outcomes (e.g., reducing the number of total laryngectomies).

This study analyzes a large multi-institutional series of 134 cases of LC T4a treated with OPHL+/− PORT, resulting in pT4a at the post-operative pathological examination and not only locally advanced at the clinical diagnosis.

Despite the limitations of a retrospective study, the goal is to better understand if and in which cases OPHL can be included among the viable options in selected pT4a LC patients who refuse a standard approach, represented by TL + PORT.

## 2. Materials and Methods

One hundred and thirty-four patients with an intermediate/locally advanced laryngeal carcinoma underwent OPHL at the Vittorio Veneto Hospital (Treviso, Italy), Martini Hospital in Turin, Candiolo Cancer Institute (FPO IRCCS—Candiolo, Italy), or the Brazilian National Cancer Institute in the period between May 1995 and February 2019.

Following the OPHL classification introduced in 2014 by the European Laryngological Society [8], all procedures were considered conventional in terms of technique, indications, and according to the ethical standards of the Institutional and/or National Research Committee and the 1964 Helsinki Declaration and its later amendments. Ethical review and approval were not required for this study in accordance with national and institutional requirements. Before surgery, all patients signed an informed consent form to disclose appropriate personal data for scientific purposes. All patients underwent the same clinical assessment within three weeks before surgery, including clinical examination, nutritional status evaluation (body mass index, BMI), biopsy/pathological examination, maxillofacial and neck MRI or CT scan (in Martini Hospital and Candiolo Cancer Institute all patients performed neck MRI; in Vittorio Veneto Hospital and the Brazilian National Cancer Institute all patients underwent neck CT-scan) and were discussed in the institutional tumor board.

Each patient provided their informed consent, including sections on laryngeal anatomy and physiology, surgical aims and indications, alternatives to surgery, complications, and physiology of the operated larynx. The Consent Form is written in a “modular” way: the surgeon defines the precise extension of the lesion, chooses the best OPHL procedure, and highlights all expected specific extensions, including total laryngectomy [17].

In cases at risk of or with limited extra-laryngeal extension through the cartilages or membranes, based on the realistic possibility of obtaining a radical resection using an OPHL, the indication for the latter was extended; it was also extended to cT4a cases in patients who refused total laryngectomy.

Pre-operative endoscopy was necessary to determine the suspicion of extra laryngeal extension based on the involved sites and to stratify cases according to T subcategories into anterior/posterior [15,18] associated with normal arytenoid motility/fixation.

MRI was particularly useful in assessing cartilage infiltration, involvement of pre-laryngeal soft tissue, pre-epiglottic space (PES), and/or paraglottic (PGS) spaces, and extra laryngeal spread.

Generally, the pre-operative selection of patients was based on the following factors: only subjects aged 70 years or less and without serious comorbidities were considered eligible for surgery [17]. Only in a few patients in excellent general condition and who were highly motivated was the age cut-off violated. Patients with severe chronic obstructive pulmonary diseases (inability to walk up two flights of stairs), severe diabetes mellitus, neurologic disorders affecting the ability to expectorate and/or swallow, or severe cardiac diseases were not considered candidates for upfront conservative surgery.

Intraoperative endoscopy allowed for a thorough reassessment of the tumor extension and highlighted some clinical-endoscopic elements of suspected initial extra laryngeal extension (Figure 1 and Figure 2).

Based on clinical work-up and subsequent pathological reports, cases were stratified into subcategories based on laryngeal compartmentalization, using a vertical virtual plane passing between the vocal process of the arytenoid and the corresponding thyroid cartilage lamina [15].

In particular, cases were assigned to either subcategory III (supraglottic/glottic/subglottic pT4a, involving the anterior laryngeal compartment, with extra laryngeal extension—through the thyrohyoid membrane, thyroid cartilage, and/or cricothyroid membrane—but with normal arytenoid mobility) or subcategory IV (supraglottic/subglottic pT4a, involving the posterior laryngeal compartment, with extra laryngeal extension—through or around the posterior portion of the thyroid lamina, through the lateral cricothyroid membrane, the cricoid cartilage and/or at the level of the cricothyroarytenoid space—and with reduced or absent arytenoid mobility).

### 2.1. Surgery

Based on the extent of the tumor, type I OPHL was reserved for tumors of the supraglottic site with possible limited extension to adjacent sites, OPHL type II was used for tumors of the glottic site with possible posterior extension to the arytenoid or superior extension to the supraglottic site, and type III OPHL was used for glottic site tumors with anterior and/or posterior subglottic extension in the absence or with minimal lymph node involvement (cN0-1). In OPHL type II, the resection involves the entire thyroid cartilage. The upper edge of the cricoid ring represents the inferior limit, whereas, in OPHL type III, the resection is extended downward to one hemi-cricoid. The clinic/pathologic feature most often characterizes the tumors amenable to OPHL type III resection is vocal cord and arytenoid fixation with cricoarytenoid joint and cricothyroid space involvement, combined with arytenoid and/or cricoid sclerosis. In this case, choosing an OPHL type II procedure would result in a greater risk of positive margins.

The suprahyoid part of the epiglottis and both cricoarytenoid units can also be resected; on this basis, according to the classification proposed by the European Laryngological Society, OPHL type II and III can be distinguished in:-OPHL type IIa and OPHL type IIIa: both cricoarytenoid units and the suprahyoid portion of epiglottis are preserved;-OPHL type IIa + ARY and OPHL type IIIa + CAU: suprahyoid portion of the epiglottis is preserved, and the resection is extended to one arytenoid (type IIa) or cricoarytenoid unit (type IIIa);-OPHL type IIb and OPHL type IIIb: both cricoarytenoid units are preserved, and the resection is extended to the whole epiglottis;-OPHL type IIb + ARY and OPHL type IIIb + CAU: the resection is extended to the entire epiglottis and one arytenoid (type IIb) or cricoarytenoid unit (type IIIb).

Most cases, particularly those at risk of being upstaged to pT4a or showing limited extra-laryngeal spreading, underwent resection of the pre-laryngeal muscles and thyroid gland (isthmus ± ipsilateral lobe), together with the dissection of the medial compartment of the neck. The resection margins were examined intraoperatively in all patients by frozen sections: if positive, the resection was extended until the margins were negative. The surgical margins were always rechecked on the final histopathological examination. This strategy, especially in pre-treated patients, helps to reduce but does not eliminate the risk of having positive margins by definitive pathology.

### 2.2. Adjuvant Treatments

Following current guidelines, the addition of adjuvant radio(chemo)therapy was discussed at the tumor board in case of gross extra-laryngeal extension, positive margins, metastatic lymph node involvement > pN1 +/− extra-nodal extension, and presence of risk features at the final pathology report.

## 3. Statistical Methods

Overall survival (OS), disease-free survival (DFS), disease-specific survival (DSS), freedom from laryngectomy (FFL), and laryngo-esophageal dysfunction-free survivals (LEDFS) were assessed by Kaplan–Meier curves. Log-rank (LR) and Gehan–Breslow–Wilcoxon (GBW, for early events) tests compared Kaplan–Meier estimates among the different subcategories.

The endpoints considered were obtained as the length of time from the date of diagnosis to the date of death (OS), date of first recurrence (DFS), date of death from disease (DSS), date of salvage laryngectomy (FFL), date of gastrostomy, tracheostomy, not intelligible voice, salvage total laryngectomy, or date of death (LEDFS).

Logistic regression analyses were performed with IBM SPSS version 24, and curves were performed with GraphPad Prism version 9.1.1, with a statistical significance threshold of *p* < 0.05. They were used to investigate which factor may influence the risk associated with OS, DFS, DSS, FFL, and LEDFS. Forest plots represent the OR (odds ratio) ± 95% CI (confidence interval) obtained from univariate analysis.

## 4. Results

Patient characteristics, distribution by involved laryngeal site, the correlation between cT and pT, and any previous treatment are shown in Table 1.

Pathological staging is shown in Table 2. On pathological examination, all 134 tumors were classified as pT4a.

Arytenoid motility was normal in 108 patients (80.6) and absent in 26 patients (19.4%).

Forty-seven patients (35.1%) classified cT2 and 58 cases (43.3%) classified cT3 resulted in pT4a on the final pathological report.

Thirty-five cases (26.1%) originated from the supraglottic site, and 99 (73.9%) showed glottic origin. In the whole cohort, 109 patients (81.3%) were N0 and 25 (18.7%) N ≥ 1.

All 134 patients underwent OPHL type I (supraglottic), II (supracricoid), or III (supratracheal), the indications and contraindications of which have been described in previous studies [10,11,13,19]. OPHLs were classified according to the European Laryngological Society Classification [8]. The classification of different surgical procedures is shown in Table 3.

One hundred and eight (80.6%) were classified as anterior, and 26 (30.5%) were classified as posterior according to subcategory stratification [14]. Resection of the arytenoid was performed in 57 cases (42.5%), while the entire cricoarytenoid unit was resected in 24 patients (17.9%). Both arytenoids were spared in the remaining 53 cases (39.6%).

One hundred and nine patients (81.3%) resulted in pN0. Twenty-five patients (18.7%) showed lymph node metastases: 10 (40.0%) pN1, 9 (36.0%) pN2, and 6 (24.0%) patients with extracapsular extension were staged as pN3b following the 8th TNM classification [6].

Neck dissection (ND), classified according to the American Academy of Otolaryngology—Head and Neck Surgery Foundation classification [20], was performed in 130 patients (97.0%), unilaterally in 78 (60.0%), and bilaterally in 52 cases (40.0%). In 72 patients (55.4%), the dissection of level VI or ipsilateral tracheoesophageal groove was also performed. In 4 patients (3.0%) staged cN0, ND was not performed.

Based on pathological findings (pN+ and/or extracapsular spread, extra-laryngeal extension, positive margins), 34 patients (25.4%) underwent adjuvant radiotherapy (RT).

The main indications for radiotherapy were: pN+ (14 patients, 10.4%), significant extra laryngeal extension (3 patients, 2.2%), and positive margins (2 patients, 1.5%).

Chemotherapy (CT) was added in 10 (7.5%) patients according to the following schedule: 100 mg/m^2^ cis-platinum on days 1, 22, and 43, concomitant with radiotherapy due to the high risk of loco-regional recurrences (pN+ Delphian lymph node; pN ≥ 1 with extra-nodal extension and pT4a with gross extra-laryngeal extension with positive/minimal margins towards pre-laryngeal soft tissues). Only one patient (0.7%) underwent CT alone.

The pathological report showed complete resection in 118 cases (88.1%); margins were negative in 79 patients (59.0%), close (less than 2 mm on specimens) in 39 cases (29.1%), while positive margins (negative on extemporaneous frozen examination, but positive on definitive histopathological examination) were found in 16 patients (11.9%).

Fifty-four cases (40.3%) also had vascular permeation, 40 patients (28.9%) had perineural invasion, and 14 (10.4%) had metastases to the Delphian lymph node.

Comparing the clinical and pathological stages showed pT and pN up-staging in 105 cases (78.4%) and 16 patients (11.9%), respectively. Conversely, pN down-staging was observed in 5 cases (3.7%).

Furthermore, among patients correctly staged cT4a → pT4a, 18 of 29 (62.1%) had a recurrence, whereas only 16 of 105 patients (15.2%) among those clinically cT2-3 → pT4a developed a recurrence (*p* < 0.001).

Twenty-one patients (15.7%) underwent completion of TL due to oncological or dysfunctional reasons.

During follow-up, 38 patients (28.4%) experienced a recurrence: 15 (11.2%) local, 11 (8.2%) regional (of these, one also had distant metastases), 7 (5.2%) loco-regional (of these, 4 also had distant metastases), and 5 (3.7%) distant metastases only.

The number of recurrences was distributed between the two subcategories: 29 of 108 (26.8%) in the anterior subcategory and 9 of 26 (36.6%) in the posterior subcategory (*p* = 0.428).

Among patients who had a recurrence, 23 (60.5%) underwent salvage surgery (total laryngectomy and/or ND); 10 (26.3%) non-surgical treatment; 2 (5.3%) RT alone; 3 (7.9%) chemotherapy; and 3 (7.9%) concurrent chemoradiation therapy (CCRT).

Five-year data on OS, DSS, DFS, FFL, and LEDFS are reported in Table 4 for the overall population, comparing anterior and posterior locations, type of surgery, patients undergoing or not undergoing adjuvant therapy, clinically staged T4a, and clinically downstaged patients.

Overall, complications (Table 5) were observed in 22 cases (16.4%): cervical bleeding in 9 patients (6.7%), and aspiration pneumonia in 5 cases (3.7%).

No patients died in the postoperative period.

Sequelae (Table 6) were observed in 28 patients (20.9%).

Twenty-two patients (16.4%) maintained the tracheostomy for a long time due to airway stenosis and were subsequently treated with CO_2_ laser resection. Of these, 18 (13.4%) were permanently decannulated.

Despite an intensive swallowing rehabilitation protocol, 13 patients (7.3%) complained of persistent dysphagia.

At the last follow-up, 89 patients (66.4%) were NED (non-evidence disease), 17 (12.7%) DWD (died with disease), 20 (14.9%) DOD (died of other diseases), and 5 (3.7%) were LWD (living with disease). Three patients (2.2%) were lost to follow-up.

The following graph shows values for OS (Figure 3), DSS (Figure 4), DFS (Figure 5), FFL (Figure 6), and LEDFS (Figure 7) for different subcategories.

## 5. Discussion

Locally advanced laryngeal cancer represents a heterogeneous group of oncologic conditions depending on the site and the extent of the tumor.

The therapeutic approach of T4a disease (N stage regardless) cannot disregard TL-PORT, as widely demonstrated by several studies in the literature. Radical surgery remains the gold standard, with acceptable oncological, functional, and quality-of-life outcomes.

A significant problem in any multidisciplinary tumor board is the high number of patients with cT4a or strongly suspected cT4a LC unwilling to accept total laryngectomy and permanent tracheostomy.

However, while radical surgery and postoperative radiotherapy can provide acceptable long-term oncological outcomes, larynx-preservation options usually offer lower locoregional control and poor potential for good functional outcomes if compared to TL-PORT; for these reasons, the latter should be reserved only for very select cases, i.e., patients showing smaller volume disease, minimal cartilage destruction, and adequate airway function [21]. Based on the favorable oncological results reported herein, OPHLs sometimes represent an alternative to TL in cases with similar clinic-radiological features. This hypothesis is based on the theoretical principle of surgical radicality.

Indeed, in the case of tumors with minimal extra-laryngeal extension, OPHLs can be modulated until radicality is achieved, as determined by pathological examination of the specimen.

Anteriorly, the strap muscles, isthmus, and sometimes the corresponding thyroid lobe are resected in principle. Posteriorly, the resection may be extended to a complete cricoid-arytenoid unit (hemicricoid plate + arytenoid). Despite these precautions, since these are locally advanced tumors, and considering the pattern of spread, the rate of positive margins is high, about 10%.

In the literature, some authors reported pT4a cases successfully treated with OPHLs [11,15,22,23,24,25,26]. These are mainly tumors with minimal extra-laryngeal extension, often staged cT3 and then found to be pT4a on final pathology examination.

Based on these observations, which are certainly not systematic and do not provide a high level of evidence in favor of conservative surgery even in cases of categorical refusal of TL, we decided to retrospectively assess outcomes of a multi-institutional cohort of 134 pT4a patients. To our knowledge, this represents the largest cohort treated with OPHL in the literature. The aim was to evaluate this surgery’s real potential in managing laryngeal T4a to favor its consideration in future guidelines.

Based on the 5-year actuarial survival rates of 82.6% and 80.1% for anterior and posterior tumor spread, respectively, OPHL can be considered a feasible treatment option following careful selection of both patients (refusing TL) and tumors (low volume cT4a–cN0, maximum cN1).

At the same time, although inferior to laryngeal preservation strategies in terms of quality of voice and swallowing, OPHLs provide competitive functional results. In about 7 of 10 cases, patients do not complain of significant laryngo-esophageal dysfunction at five years, and 8 of 10 still maintain their larynx.

These data are globally in line with results in the literature [11,14,15,19,27,28,29].

A multidisciplinary evaluation in a referral center for laryngeal cancer treatment, as well as an accurate diagnostic procedure resulting from a strong collaboration with radiologists, are the keys to achieving good functional outcomes and minimizing the risk of developing recurrences and the consequent need for salvage laryngectomy, even in the limited number of T4a LC patients eligible for OPHL who categorically refused TL (in our experience <10% of cases) [30].

In this retrospective series, 78.4% of the cases were found to be pT4a following pathological analysis on the specimen while they were erroneously understaged (cT2/cT3) during the workup.

This aspect can certainly be considered an index of poor workup. Still, it can often result from interpretative difficulties of the clinical and radiological features of the disease, which also occur in high-volume centers [21].

In our experience, this was observed mainly for anterior, low-volume tumors with preserved arytenoid motility.

As shown in logistic regression analysis, clinical T and N stages represent the only parameters that negatively affect survival in this case series. At the same time, tumor spread (anteriorly versus posteriorly) did not significantly influence either prognosis or oncological/functional outcomes. Similar results, albeit in a smaller cohort of T4a, all treated with TL +/− PORT, were reported by Marchi and colleagues [31].

cT4a tumors are more aggressive than cT2-T3 tumors and must be considered “true” loco-regional diseases. They are more difficult to control using any laryngeal preservation approaches due to the extensive involvement of the laryngeal framework and extra-laryngeal anatomical structures. As evidence of this, in correctly staged cT4a cases characterized by bulky disease, there was extensive destruction/invasion of the framework and cricothyroid membrane; additionally, the results of OPHLs in this series proved to be poor, with OS of 68.2% and DFS of 49.5 at five years.

In contrast, cases that were incorrectly staged cT2-T3 tumors and then upstaged as pT4a still showed a predominantly endo-laryngeal disease, with minimal extra-laryngeal extension proven by pathological examination. In fact, in the current series, we are mainly dealing with cases selected for age, absence of comorbidity, and T stage (pT4a with extra laryngeal extension, tending to be anterior, in 78.4% of cases staged cT2/T3 at the end of the diagnostic work-up) and predominantly without clinically involved lymph nodes (mainly cN0 up to cN1, without clinical evidence of clinical/radiological extra-nodal extension).

Considering the obtained results, the oncological outcomes in these “early” pT4a cases are stable and comparable to those described in other cohorts, with most cases of intermediate T stages, meaning that clinical extension and tumor dimension are fundamental parameters for a correct indication for OPHL. Similar, well-selected cases, also from retrospective series from single institutions, have shown promising results with nonsurgical larynx-preserving approaches [21,32].

In our opinion, the results of this series provide valuable data to explain the worse oncological outcomes of bulky cT4a tumors treated with any conservative approach to the larynx, both surgery and chemoradiation [32,33].

Another interesting aspect is that only 34 patients (25.4%) underwent adjuvant (chemo)radiotherapy (pN+ especially level VI and/or extracapsular spread: 14 patients (10.4%), +/− important extra-laryngeal extension: 3 patients (2.2%), +/− extensive positive margins: 2 patients (1.5%)). These are undoubtedly the cases with the worst prognosis, as shown by oncological and functional parameters, but they must also be considered cases in which the philosophy of surgical organ and function preservation with a single therapy modality has failed.

Considering these data, an even greater effort should be made to reduce the percentage (close to zero) of cases in which the tumor board deems it safer to give an indication for adjuvant therapy after surgery.

This goal can be achieved with an even greater selection of the T and N parameters by carrying out the CT-scan and the MRI in the work-up.

Overall, in the remaining 74.6% of cases, the tumor boards (three different centers) decided not to indicate adjuvant RT, despite having an extra-laryngeal extension, considering the obtained radicality as sufficient to achieve loco-regional control and keeping in mind the high risk of toxicity/functional damage related to adjuvant treatments. Oncological and functional results have undoubtedly favored this choice, even if it can be considered questionable since the overall risk of recurrence in the present case series is >25%. This strategy should be the subject of further investigation as it represents a choice that, theoretically, based on a dimensional criterion and not on the pathological stage, violates the principle that in pT4a there is a clear indication for postoperative radiotherapy.

This strategy also needs a higher level of standardization since it must be accepted that the indication for adjuvant therapy should meet clear and more repeatable criteria, both in terms of indication and radiation ranges.

Muscatello et al. found that OPHL types II and III may have favorable oncological and functional results even with adjuvant (chemo-)radiotherapy. However, it remains challenging to define a standardized protocol since treatment for advanced LC must be wisely tailored based on careful patient selection [34].

An important aspect is related to complications and sequelae. Since these were pT4a cases, a significant number of patients (35, 26.1%) underwent type III OPHL, which is known to have a higher rate of sequelae, particularly stenotic sequelae of the neoglottis. Nevertheless, OPHL type III versus CCRT protocols is viable in selected pT4a LC in terms of prognostic and functional outcomes and in reducing the number of salvage total laryngectomies [14].

Overall, complications were observed in 22 cases (16.4%), but no patient died in the post-operative period.

Twenty-two patients (16.4%) maintained the tracheostoma because of airway stenosis, which was subsequently treated with one or more CO_2_ laser resections. Of these, 18 (13.4%) were permanently decannulated.

Focusing on the functional aspect, as already reported in previous observations, LEDFS, considered the most composite functional endpoint, did not demonstrate significant differences between type II and type III OPHLs.

These data show that in centers where high volumes of OPHL are managed (type III OPHLs included), good expertise is needed in so-called endoscopic remodeling surgery by CO_2_ laser and injection laryngoplasty to reduce/abolish relevant dysphagic and dysphonic sequelae. Equal emphasis must be placed on follow-up as recurrence on T often occurs outside the larynx, with a longer latency than in other partial laryngectomy procedures.

Twenty-one patients (15.7%) underwent total laryngectomy for oncological or dysfunctional reasons.

Laryngectomy-free survival, especially in the most favorable cases with minimal extra-laryngeal extension, is very high and in line with data from patients undergoing the same operation for lower T-stage tumors. This allows the stating of the OPHL validity for selected cases of borderline T3, at high risk of proving to be a true pT4a.

This means that, despite the best efforts to achieve a correct selection of patient and tumor, even if the pathological report reveals a locally more advanced tumor (especially in terms of T stage rather than N stage), it ultimately makes it possible to successfully spare laryngeal functions through OPHL, which is associated with good oncological results, even if this requires the surgical sacrifice of a substantial portion of laryngeal and prelaryngeal tissues.

This is also made possible due to a modular approach strategy that allows intra-operative widening of the resection according to the actual extent of the tumor. In this way, it is possible to add significant margins to reach effective radicality to a planned resection that, in the end, would prove too risky to be insufficient. Nevertheless, the 11.9% positive margins on the final pathology report require the utmost caution in indications, especially in recurrences after radiotherapy and laser surgery.

Future multi-institutional prospective studies are necessary to verify the data of this retrospective study, which refer to the activity of three referral centers that differ in terms of background and organization, but not in terms of the experience of the operators.

## 6. Conclusions

In conclusion, the results of this large series, although conditioned by the retrospective approach along a period of about 24 years, support the observations already reported by several authors and highlight the good onco-functional results of low volume, anterior pT4a laryngeal tumors with minimal extra-laryngeal extension treated with OPHLs. Therefore, such a surgical strategy should no longer be considered anecdotal.

The level of standardization of the indication for OPHL, especially in cT3 laryngeal tumors with a high risk of turning out to be pT4a and in safe cases of small anterior cT4a, based on the criteria of laryngeal compartmentalization, should allow consideration of the technique as a valid therapeutic option in case of the patient’s absolute refusal of total laryngectomy or non-surgical protocols with concomitant chemo-radiotherapy.

Extending the limit of resection and including a large part of the cricoid cartilage and one crico-arytenoid unit (type III OPHL + CAU) expanded the indications of type II OPHL + ARY, allowing for a safer resection of advanced and challenging laryngeal tumors.

## Figures and Tables

**Figure 1 cancers-15-02861-f001:**
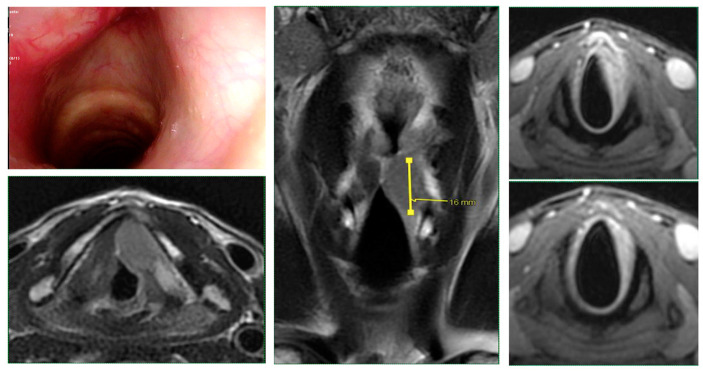
Glottic squamous cell carcinoma cT3 (left vocal cord): endoscopic and radiological (MRI) picture: the tumor was likely to be sub-staged with endoscopic evaluation alone.

**Figure 2 cancers-15-02861-f002:**
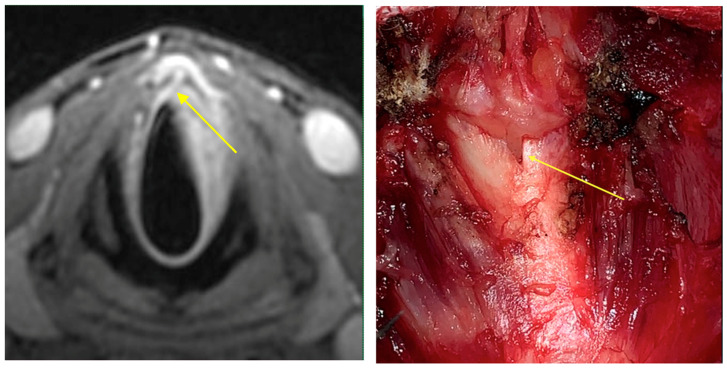
The radiological finding of suspected extra laryngeal extension through the cricothyroid membrane is evidenced (yellow arrow). Intraoperative evaluation (yellow arrow) confirmed the extra laryngeal spread through a vascular foramen of the cricothyroid membrane.

**Figure 3 cancers-15-02861-f003:**
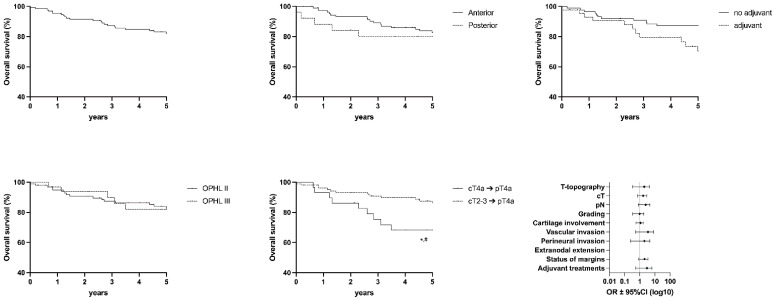
Overall Survival: Forrest Plot and Kaplan–Meier Estimator. Kaplan–Meier: * = *p* < 0.05 (LR); # = *p* < 0.05 (GBW).

**Figure 4 cancers-15-02861-f004:**
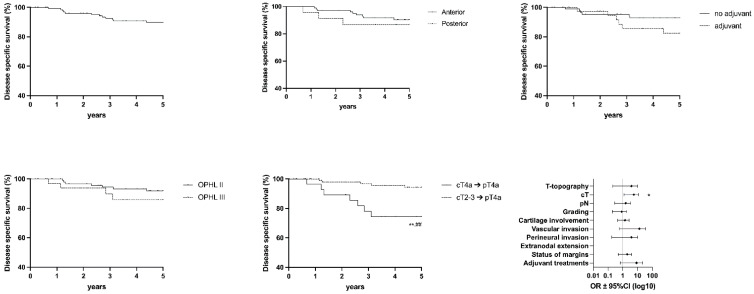
Disease Specific Survival: Forrest Plot and Kaplan–Meier Estimator. Kaplan–Meier: ** = *p* < 0.01 (LR); ## = *p* < 0.01 (GBW). Forrest Plot: * = <0.05.

**Figure 5 cancers-15-02861-f005:**
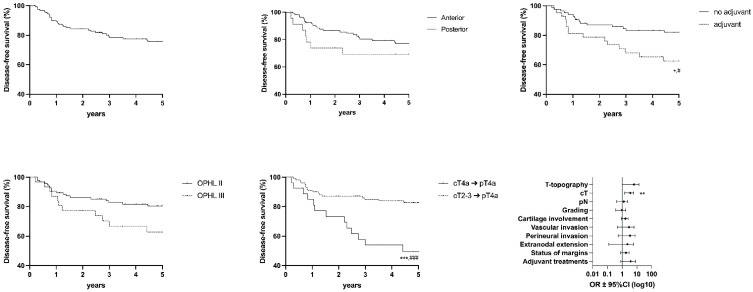
Disease-free Survival: Forrest Plot and Kaplan–Meier Estimator. Kaplan–Meier: * = *p* < 0.05, *** = *p* < 0.001 (LR); # = *p* < 0.05, ### = *p* < 0.001 (GBW). Forrest Plot: ** = <0.01.

**Figure 6 cancers-15-02861-f006:**
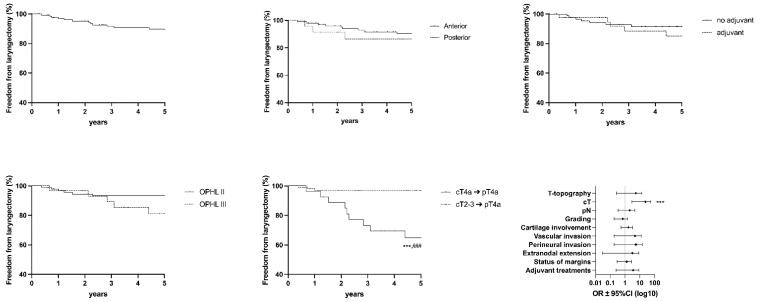
Freedom From Laryngectomy: Forrest Plot and Kaplan–Meier Estimator. Kaplan–Meier: *** = *p* < 0.001 (LR); ### = *p* < 0.001 (GBW). Forrest Plot: *** = <0.001.

**Figure 7 cancers-15-02861-f007:**
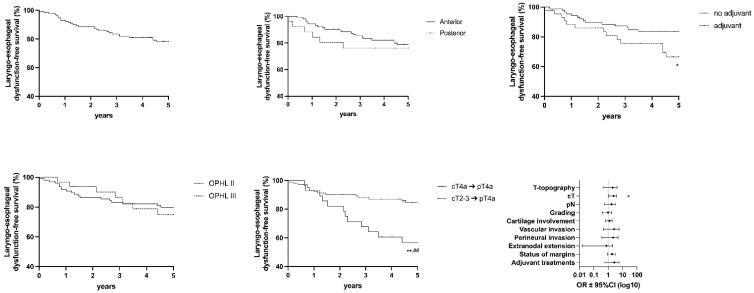
Laryngoesophageal Dysfunction Free Survival: Forrest Plot and Kaplan–Meier Estimator. Kaplan–Meier: * = *p* < 0.05, ** = *p* < 0.01 (LR); ## = *p* < 0.01 (GBW). Forrest Plot: * = <0.05.

**Table 1 cancers-15-02861-t001:** Demographic data—134 patients.

		No. of Patients (%)
Age	Mean ± standard deviation	61.5 ± 9.5
	Range	41–90
Gender	Male	121 (90.3%)
	Female	13 (9.7%)
Arytenoid mobility	Normal	108 (80.6%)
	Impaired/fixed	26 (19.4%)
cT vs. pT	cT2-3 → pT4a	105 (78.4%)
	cT4a → pT4a	29 (21.6%)

**Table 2 cancers-15-02861-t002:** Pathological staging—134 patients.

pTN		Glottic	Supraglottic
pT4	N0	85 (85.9%)	24 (68.6%)
	N1	7 (7.1%)	3 (8.6%)
	N2	3 (3.0%)	6 (17.1%)
	N3b	4 (4.0%)	2 (5.7%)
Total		99 (73.9%)	35 (26.1%)

**Table 3 cancers-15-02861-t003:** Surgeries carried out on the 134 patients included in the study.

Type of Treatment (OPHL)	N (%)	cT2	cT3	cT4a
I + BOT	2 (1.5%)	-	1	1
IIa	29 (21.6%)	15	12	2
IIa + ARY	31 (23.1%)	13	13	5
IIb	14 (10.4%)	3	6	5
IIb + ARY	26 (19.4%)	10	15	1
IIIa	5 (3.7%)	2	2	1
IIIa + CAU	21 (15.7%)	2	7	12
IIIb	3 (2.2%)	1	-	2
IIIb + CAU	3 (3.4%)	1	2	-

BOT: resection extended to the base of the tongue; ARY: resection extended to one arytenoid; CAU: resection extended to one cricoarytenoid unit.

**Table 4 cancers-15-02861-t004:** Kaplan–Meier estimates, ±95% confidence interval.

	**Global**		**Subcategory**				**Adjuvant Treatment**			
			**Anterior**		**Posterior**		**Adjuvant**		**No Adjuvant**	
OS	82.1	(74.1–87.9)	82.6	(73.5–88.8)	80.1	(68.5–91.2)	70.4	(52.6–82.5)	87.2	(78.1–92.7)
DSS	89.8	(82.6–94.1)	90.4	(82.3–94.9)	86.7	(64.3–95.5)	82.5	(65.0–91.7)	92.8	(84.6–96.7)
DFS	75.7	(67.1–82.4)	77.0	(67.4–84.1)	69.3	(46.1–84.0)	62.5	(45.4–75.6)	82.0	(71.9–88.7)
FFL	89.7	(82.6–94.1)	90.4	(82.3–94.9)	86.5	(63.7–95.4)	85.1	(67.7–93.6)	91.6	(83.2–95.9)
LEDfs	78.3	(70.0–84.6)	78.9	(69.4–85.7)	76.0	(54.2–88.5)	66.5	(49.1–79.1)	83.6	(73.9–90.0)
			**Surgery**				**Staging**			
			**OPHL II**		**OPHL III**		**staged cT4a**		**Clinically** **understaged**	
OS			82.8	(73.4–89.1)	82.0	(61.8–92.2)	68.2	(47.8–82.1)	86.3	(77.6–91.8)
DSS			91.9	(83.8–96.1)	85.9	(66.5–94.5)	74.4	(53.7–86.9)	94.4	(87.0–97.6)
DFS			80.5	(70.8–87.3)	62.8	(42.7–77.5)	49.5	(29.1–66.9)	82.7	(73.6–88.9)
FFL			93.3	(85.7–96.9)	81.1	(60.2–91.7)	64.8	(43.1–80.0)	97.0	(90.9–99.0)
LEDfs			79.8	(70.1–86.6)	74.9	(54.3–87.3)	56.7	(36.4–72.7)	84.7	(75.9–90.5)

Kaplan–Meier estimates: % ± (95%CI); GBW: Gehan–Breslow–Wilcoxon test for early events; LR: Log Rank (Mantel–Cox) test for 5-year differences.

**Table 5 cancers-15-02861-t005:** Complications in the 134 patients included in the study.

Complications	Number of Events (%)	Adjuvant Therapy (N = 45)	Non-Adjuvant Therapy (N = 89)	cT2-3 (105)	cT4 (29)
Cervical bleeding	9 (6.7%)	6 (13.3%)	3 (3.4%)	9 (7.4%)	0 (0.0%)
Kidney failure	1 (0.7%)	0 (0.0%)	1 (1.1%)	1 (1.0%)	0 (0.0%)
Sepsis	0 (0.0%)	0 (0.0%)	0 (0.0%)	0 (0.0%)	0 (0.0%)
Acute myocardial infarction	1 (0.7%)	0 (0.0%)	1 (1.1%)	1 (1.0%)	0 (0.0%)
Respiratory failure	0 (0.0%)	0 (0.0%)	0 (0.0%)	0 (0.0%)	0 (0.0%)
Aspiration pneumonia	5 (3.7%)	3 (6.7%)	2 (2.2%)	5 (4.8%)	0 (0.0%)
Stroke	0 (0.0%)	0 (0.0%)	0 (0.0%)	0 (0.0%)	0 (0.0%)
Wound infection	1 (0.7%)	1 (2.2%)	0 (0.0%)	1 (1.0%)	0 (0.0%)
Fistula	2 (1.5%)	2 (4.4%)	0 (0.0%)	2 (1.9%)	0 (0.0%)
Other	3 (2.2%)	2 (4.4%)	1 (1.1%)	3 (2.8%)	0 (0.0%)
Total	22 (16.4%)	14 (31.1%)	8 (9.0%) **	22 (21.0%)	0 (0.0%) *

*: *p* < 0.05; **: *p* < 0.01.

**Table 6 cancers-15-02861-t006:** Sequelae in the 134 patients included in the study.

Sequelae	Number of Events (%)	Adjuvant (N = 45)	Non-Adjuvant (N = 89)	cT2-3 (105)	cT4 (29)
Tracheostoma stenosis	3 (2.2%)	0 (0.0%)	3 (3.4%)	3 (2.8%)	0 (0.0%)
Laryngeal tight stenosis	18 (13.4%)	8 (17.8%)	10 (11.2%)	15 (14.3%)	3 (10.3%)
Dysphagia	5 (3.7%)	4 (8.9%)	1 (1.1%)	4 (3.8%)	1 (3.4%)
Laryngeal tight stenosis + Tracheostoma stenosis	0 (0.0%)	0 (0.0%)	0 (0.0%)	0 (0.0%)	0 (0.0%)
Dysphagia + Laryngeal tight stenosis	2 (1.5%)	0 (0.0%)	2 (2.2%)	1 (1.0%)	1 (3.4%)
Total	28 (20.9%)	12 (26.7%)	16 (18.0%)	23 (21.9%)	5 (17.2%)

## Data Availability

The Authors clarify that since it is a retrospective series over a period of 25 years and since the laws on privacy and consent have changed in the different national realities, the sharing of data outside the individual institutions is not possible as it is not expressly requested of all patients, some of whom have already died.

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
