# Peer review of "Partial Laryngectomy for pT4a Laryngeal Cancer: Outcomes and Limits in Selected Cases"

_cancers, 2023, doi:10.3390/cancers15102861_

Round 1
Reviewer 1 Report (Previous Reviewer 1)
I have previously reviewed this manuscrpt and I confirm that there are some criticisms as yet specified. Particularly , the reported data did'nt suggest to consider partial laryngectomy for pT4a laryngeal cancer, considering the survival difference between T3 and T4 clinical staged. Further , should be interesting in the discussion to report comparative data with pT4 laryngeal cancer treated by organ preservation protocols.
Author Response
We thank the reviewer for the frank criticism regarding the indication for partial laryngectomy in pT4a laryngeal cancer.
In the present review only the naive were considered and analyzed.
As suggested, in the discussion cT4a cases treated with organ preservation protocols and OPHL were compared.
As a result of this comparison more balanced conclusions emerged and for this we are grateful to the reviewer for the suggestions.
Reviewer 2 Report (Previous Reviewer 2)
I have no further comments to add than the previous review.
Author Response
We thank the reviewer.
In the present review only the naive were considered and analyzed.
In the discussion cT4a cases treated with organ preservation protocols and OPHL were compared.
Reviewer 3 Report (Previous Reviewer 3)
The authors improved the manuscript. However, an issue remain: patients who had previous treatments must be analyzed separately. Stratification for other risk factors (staging, anterior/posterior subcategory, adjuvant therapy) should be performed after excluding patients with previous treatments.
Author Response
We thank the reviewer for the frank criticism regarding the pretreated patients.
In the present review only the naive were considered and analyzed.
In the discussion cT4a cases treated with organ preservation protocols and OPHL were compared.
As a result of this comparison more balanced conclusions emerged and for this we are grateful to the reviewer for the suggestions.
Reviewer 4 Report (New Reviewer)
The authors present a paper in which they attempt to determine whether there is a role of partial surgery in the management of T4 laryngeal tumors. T4a laryngeal cancers represent a heterogeneous group of lesions amenable to partial and total laryngectomies, as well as non-surgical therapeutic strategies. In advanced stages of primary disease (T3 and T4 tumours), the only recommended option for the preservation of the larynx is chemoradiotherapy, with the exception under some ASCO guidelines, of T3 supraglottic tumours with invasion limited to the pre-epiglottic space, in which organ-conserving surgery is presented as a viable option. From these guidelines, it could be thought that, if the chosen treatment is surgery, all T3 and T4 laryngeal tumours would require a total laryngectomy. However, there are several studies that show that organ-conserving surgery is an option in certain cases. Moreover, T classification is insufficient to properly evaluate which surgical options for organ preservation can be applied to a given tumour. Therefore, there is a wide range of function-preserving surgical techniques available, leading to excellent oncological results, as well as excellent functional results in terms of speech and swallowing, and with good comparison with the results from non-surgical functional preservation protocols.
Succo et al present their experience in a large series of patients and advocate the application of compartmentalised surgery in small T4 laryngeal tumours. However, the authors describe T4 tumours but in reality there are initially only 41 cT4 tumours. Subsequent study increases the tumour stage. Therefore, it cannot be concluded that T4 tumours can be treated with a partial approach. This reflects a poor preoperative work-up or reflects the difficulty of staging. Furthermore, these are patients who refuse total laryngectomy. It would be better to change the focus of the work because under ideal conditions of perfect staging, few T4 tumours are candidates for partial surgery and the conclusion of this work is that many would be.
In the series presented it seems clear that many of the patients benefit from partial surgery; however up to 30% of patients required adjuvant RT and 20% chemotherapy. This implies that one third of the patients were treated with more than one type of treatment, which increases toxicity. These patients could have been treated directly with non-surgical treatment. This point should be discussed.
The oncological results obtained are appropriate and comparable to other series. Emphasis should therefore be placed on the functional results. The authors should better point out how they evaluate swallowing in the patients and should better specify the functional outcomes, as this is not clear. It would be interesting to look at swallowing-related quality of life using questionnaires such as the MDADI.
Supraglottic laryngectomy is known to have optimal oncological and functional results. Similarly, supracricoid laryngectomy, in expert hands, also achieves good results. However, supratracheal laryngectomy (even supracricoid laryngectomy with a single functional cricoarytenoid unit) is challenging and very few authors are able to obtain acceptable functional results. Therefore, I would like to know the functional results separately, according to the various techniques.
The term “small T4 tumours” needs to be clarified.
The graphics should be improved, they are uncomfortable to read.
Table 4 is very verbose and complex. It should be simplified or made more user-friendly.
In conclusion, the work is correct from my point of view but I think that some aspects need to be improved.
Author Response
We thank the reviewer for the frank criticism regarding the indication for partial laryngectomy in pT4a laryngeal cancer and the too much emphasis given to the undoubtedly good results obtained in a cohort essentially composed by highly selected patients.
In the present review only the naive were considered and analyzed.
As suggested, in the discussion cT4a cases treated with organ preservation protocols and OPHL were compared.
As a result of this comparison more balanced conclusions emerged and for this we are grateful to the reviewer for the suggestions.
The term small T4 tumors was substituted with low volume T4a tumors.
As suggested table 4 was simplified.
Reviewer 5 Report (New Reviewer)
This is a well-performed study and paper. I would still ask the authors to elaborate more the problem of classifying T2 tumors. These form a challenging group of laryngeal cancers since the vocal cord mobility (fixed arytenoid or not?) is hard to define preoperatively. In the present study 136 cases were upstaged from cT2-3 to pT4a. In total, 75 of them were cT2. In my opinion, this problem area is not discussed sufficiently in the paper.
Author Response
We thank the reviewer for the frank criticism regarding the problem of clinical downstaging detected in the present study.
In the present review only the naive were considered and analyzed.
As suggested, in the discussion the problem of incorrect work up has been addressed. This is an issue currently encountered even in Centers highly specialized for the treatment of laryngeal cancer.
We agree with the reviewer that maximum efforts must be made to minimize the number of incorrectly staged cases at the work up.
As a result more balanced conclusions emerged and for this we are grateful to the reviewer for the suggestions.
Round 2
Reviewer 1 Report (Previous Reviewer 1)
I have no more comments.
Reviewer 3 Report (Previous Reviewer 3)
Thank you for improving the manuscript.
Reviewer 4 Report (New Reviewer)
A aigree with this version
This manuscript is a resubmission of an earlier submission. The following is a list of the peer review reports and author responses from that submission.
Round 1
Reviewer 1 Report
The study “Partial laryngectomy for pT4 laryngeal cancer: outcomes and limits in selected cases” in its intent is interesting , however many questions arise.
Introduction
At page 2 line 98 the authors should clarify that the study analyze al large multi-institutional series of 177 cases of LC treated with OPHL +/-PORT resulted pT4 at the post-operative pathological examination and not only locally advanced at clinical diagnosis.
At page 2 line 51 I suggest to add the following reference:
Allegra, Eugenia, et al. "Role of clinical-demographic data in survival rates of advanced laryngeal cancer." Medicina 57.3 (2021): 267.
Materials and Methods
At page 3 line 104, the authors report “One hundred seventy-seven patients with advanced laryngeal carcinoma underwent OPHL”. However they report that 61 (34.4%) of the included patients were classified cT2.
Because of the hight rate of T clinical understadied , all of the 76.8% of the cT2-cT3 resulted pT4, there have been maybe some criticisms in the pre-operative workup.
In which cases the authors performed MRI and CT and in which MRI or CT before surgery?
The authors should report which type of OPHL was performed in the cT2-cT3 and which in the cT4 patients.
Results
From the results only patients staged cT4 were confirmed as pT4 while all cT2-cT3 resulted to be pT4 , how they explain this data?
At page 8 the authors should report the number of recurrences distributed between cT2-cT3 and cT4 , because that difference has been reported as statistically significant (P<0.001)
In the tables 5 and 6 ,the authors should add the variable cT classification with statistical analysis.
Fron the DFS, FFL and LEDFS analysis seems to be a significant difference between patients classified cT2-cT3 and those cT4. The authors should report the statistical significance.
Oncological and functional results seems to be better for cT2-cT3 patients than cT4 , these data confirm that OPHL should be considered for cT4 laryngeal cancer only for who refuse total laryngectomy.
At table 4 the authors should add statistical analysis results (p value).
DISCUSSION
The authors should discuss about the results obtained in terms of DFS, FFL and LEDFS in cT2-cT3 compared to cT4 patients. This because the OPHL indication for laryngeal cancer should be made on clinical TNM classification and not on the possible pT classification.
The study show that OPHL obtain good oncological and functional results in the patient clinically stadied cT2-cT3 resulted pT4 post surgery.
On the other hand, it cannot be argued that in patients staged cT4 and confirmed pT4, OPHL gives better or ethically acceptable results compared to total laryngectomy, almost all of the patients were in any case subjected to total laryngectomy.
In conclusion, the study emerges the conclusion that in cT4 laryngeal carcinomas, which refuse total laryngectomy, the use of OPHL does not seem to give better results than the organ presevation strategy and in any case both give lower results than total laryngectomy. Patients who refuse total laryngectomy should be adequately informed of this.
Author Response
Comment: Introduction
At page 2 line 98 the authors should clarify that the study analyze al large multi-institutional series of 177 cases of LC treated with OPHL +/-PORT resulted pT4 at the post-operative pathological examination and not only locally advanced at clinical diagnosis.
We thank the Reviewer for the suggestion. We added the sentence “… resulted pT4 at the post-operative pathological examination and not only locally advanced at clinical diagnosis”.
Comment: At page 2 line 51 I suggest to add the following reference:
Allegra, Eugenia, et al. "Role of clinical-demographic data in survival rates of advanced laryngeal cancer." Medicina 57.3 (2021): 267.
We thank the Reviewer for the suggestion. We added the reference Allegra E, Bianco MR, Ralli M, Greco A, Angeletti D, de Vincentiis M. Role of Clinical-Demographic Data in Survival Rates of Advanced Laryngeal Cancer. Medicina (Kaunas) 2021 Mar 15;57(3):267. doi: 10.3390/medicina57030267
Comment: “Materials and Methods
At page 3 line 104, the authors report “One hundred seventy-seven patients with advanced laryngeal carcinoma underwent OPHL”. However they report that 61 (34.4%) of the included patients were classified cT2.
Because of the hight rate of T clinical understadied, all of the 76.8% of the cT2-cT3 resulted pT4, there have been maybe some criticisms in the pre-operative workup.
In which cases the authors performed MRI and CT and in which MRI or CT before surgery?
We thank the Reviewer for the comment. We added the term “intermediate” at page 3, line 104.
The article describes the results of a large multi-institutional case series of laryngeal cancer pT4a treated with OPHL +/- PORT in 3 different hospitals (two sites in two different Italian regions and one in South America (Brasil), each with different radiological diagnostic protocols).
Comment: “The authors should report which type of OPHL was performed in the cT2-cT3 and which in the cT4 patients
We thank the Reviewer for the comment. We added this information in Table 3
Comment: “Results
From the results only patients staged cT4 were confirmed as pT4 while all cT2-cT3 resulted to be pT4 how they explain this data?
We thank the Reviewer for the comment. Probably in the discussion it had not been well highlighted that, although it was a cohort of patients resulting in pT4a, the fundamental variable for the prognosis was the clinical staging. Indeed in Graphs 6 and 7, which have been added in the revised manuscript, it can be better appreciated (particularly in naive patients) how the cT2-T3 resulting in pT4a show oncological results almost identical to the cT2-T3 --> pT2- pT3
Comment: “At page 8 the authors should report the number of recurrences distributed between cT2-cT3 and cT4, because that difference has been reported as statistically significant (P<0.001)
We thank the Reviewer for the comment. At the page 8 line 295 - 297 we have reported the numbers of recurrences.
Comment: “In the tables 5 and 6,the authors should add the variable cT classification with statistical analysis.
We thank the Reviewer for the comment. We added the variable cT classification with statistical analysis in table 5 and 6
Comment: “From the DFS, FFL and LEDFS analysis seems to be a significant difference between patients classified cT2-cT3 and those cT4. The authors should report the statistical significance.
We thank the Reviewer for the comment. The statistical significance is reported in the graphs
Comment: “At table 4 the authors should add statistical analysis results (p value).
We thank the Reviewer for the comment. We have added the p value in table 4
Comment: “DISCUSSION
The authors should discuss about the results obtained in terms of DFS, FFL and LEDFS in cT2-cT3 compared to cT4 patients. This because the OPHL indication for laryngeal cancer should be made on clinical TNM classification and not on the possible pT classification.
The study show that OPHL obtain good oncological and functional results in the patient clinically stadied cT2-cT3 resulted pT4 post surgery.
On the other hand, it cannot be argued that in patients staged cT4 and confirmed pT4, OPHL gives better or ethically acceptable results compared to total laryngectomy, almost all of the patients were in any case subjected to total laryngectomy.
In conclusion, the study emerges the conclusion that in cT4 laryngeal carcinomas, which refuse total laryngectomy, the use of OPHL does not seem to give better results than the organ preservation strategy and in any case both give lower results than total laryngectomy. Patients who refuse total laryngectomy should be adequately informed of this.
We thank the Reviewer for the comment and we agree with him. In the section “Discussion” we added some sentences (lines 470-477) about the different results in terms of DFS, FFL and LEDFS obtained in cT2-cT3 compared to cT4a patients.
All patients were discussed in the Tumor Board and refused categorically total laryngectomy. Each patient signed a specific informed consent, including sections on laryngeal anatomy and physiology, surgical aims and indications, alternatives to surgery, complications, and physiology of the operated larynx. The Consent Form is written in a “modular” way: the surgeon defines the precise extension of the lesion, chooses the best OPHL procedure and highlights all possible expected extensions specific for the patient, including total laryngectomy (lines 121-126).
Reviewer 2 Report
This retrospective study focused on the onco-functional results of early pT4a laryngeal tumors (177 patients) treated with open partial horizontal laryngectomies.
- References are missing at the end of the manuscript (so, this section must to be improved).
- The visual quality of Forrest Plot and Kaplan-Meier Estimator graphics may be slightly improved, if feasible.
Author Response
Comment: “References are missing at the end of the manuscript (so, this section must to be improved).
We thank the Reviewer for the comment. We have checked the manuscript and the section “References” now is present. Probably there was an error uploading the article
Comment: “The visual quality of Forrest Plot and Kaplan-Meier Estimator graphics may be slightly improved, if feasible.
We thank the Reviewer for the comment. We tried to improve the quality of Forrest Plot and Kaplan-Meier Estimator graphs but it is not feasible.
Reviewer 3 Report
This is an interesting study about partial laryngectomy for selected pT4a laryngeal cancer. The study included 177 patients with pT4a laryngeal carcinoma treated with open partial horizontal laryngectomies (OPHL) +/- post operative radiation therapy (PORT).
The paper is well written. However, some issues remain.
I think that patients who had previous treatments must be analyzed separately (i.e., stratification for other risk factors, like staging, anterior/posterior subcategory, adjuvant therapy) should be performed after excluding patients with previous treatments).
P values should be reported in tables 5 and 6 and in the graphics.
The authors should make hypotheses about the better survival outcomes in cT2/3 that were upstaged to pT4a compared to cT4a.
References are lacking.
Author Response
Comment: “I think that patients who had previous treatments must be analyzed separately (i.e., stratification for other risk factors, like staging, anterior/posterior subcategory, adjuvant therapy) should be performed after excluding patients with previous treatments).
We thank the Reviewer for the comment. We analyzed separately patients who had previous treatments. We added Graph 6 and 7
Comment: “P values should be reported in tables 5 and 6 and in the graphics.
We thank the Reviewer for the comment. We added the p values in tables 5 and 6
Comment: “The authors should make hypotheses about the better survival outcomes in cT2/3 that were upstaged to pT4a compared to cT4a
We thank the Reviewer for the comment. We added some sentences about this aspect (lines 470-475)
Comment: “References are lacking.
We thank the Reviewer for the comment. We have checked the manuscript and the section “References” now is present. Probably there was an error uploading the article
Reviewer 4 Report
Abstract: In the sentence "The comparison between clinical and pathological stages showed pT and pN up-staging in 136 cases (76.8%) and 22 patients (12.4%), respectively." (ll. 34+35), it remains unclear what was up-staged (T stage, N stage? patients/cases?). Please clarify. Laryngo-esophageal dysfunction free survival and postoperative period (line 39, duration??) should be explained.
"Laryngeal cancer (LC) is one of the most common head and neck cancers, accounting for 2% of all malignant neoplasms and approximately 60% of those in the head and neck region, with 110,000-130,000 new cases expected annually worldwide. [1] " The epidemiological data appear wrong, most head and neck cancers occur in the oropharynx and oral cavity.
As I tried to find the source of this information, I did not find a reference list, therefore, it does not make sense to continue the review at this point.
Searching for the reference list, I found that this retrospective cohort study has no ethics approval from an institutional board or something corresponding ("Ethical review and approval were were not required for this study in accordance with national and institutional requirements.") which, in my opinion, is needed. At least, for a study like this, it would be needed in my institution.
I recommend rejecting the manuscript and encourage to re-submit with a reference list and an ethics statement.
Author Response
Comment: “Abstract: In the sentence "The comparison between clinical and pathological stages showed pT and pN up-staging in 136 cases (76.8%) and 22 patients (12.4%), respectively." (ll. 34+35), it remains unclear what was up-staged (T stage, N stage? patients/cases?). Please clarify.
We thank the Reviewer for the comment. We modified the sentence as follows “…pT up-staging in 136 cases (76.8%) and pN up-staging in 22 patients (12.4%)”.
Comment: “Laryngo-esophageal dysfunction free survival and postoperative period (line 39, duration??) should be explained.
We thank the Reviewer for the comment. We added the sentence “rate of patients surviving without a local recurrence or laryngectomy and do not have a feeding tube or a tracheostomy) …” to explain the term laryngo-esophageal dysfunction free survival (line 33-34).
Comment: “"Laryngeal cancer (LC) is one of the most common head and neck cancers, accounting for 2% of all malignant neoplasms and approximately 60% of those in the head and neck region, with 110,000-130,000 new cases expected annually worldwide. [1] " The epidemiological data appear wrong, most head and neck cancers occur in the oropharynx and oral cavity. As I tried to find the source of this information, I did not find a reference list, therefore, it does not make sense to continue the review at this point.
We thank the Reviewer for the comment. We made a typing error and we agreed with the Reviewer that most head and neck cancers occur in the oropharynx and oral cavity. We corrected the mistake. Moreover we have checked the manuscript and the section “References” now is present. Probably there was an error uploading the article
Comment: “Searching for the reference list, I found that this retrospective cohort study has no ethics approval from an institutional board or something corresponding ("Ethical review and approval were not required for this study in accordance with national and institutional requirements.") which, in my opinion, is needed. At least, for a study like this, it would be needed in my institution. I recommend rejecting the manuscript and encourage to re-submit with a reference list and an ethics statement.
We thank the Reviewer for the comment. All patients were discussed in the Tumor Board. Each patient signed a specific informed consent, including sections on laryngeal anatomy and physiology, surgical aims and indications, alternatives to surgery, complications, and physiology of the operated larynx. The Consent Form is written in a “modular” way: the surgeon defines the precise extension of the lesion, chooses the best OPHL procedure and highlights all possible expected extensions specific for the patient, including total laryngectomy (lines 121-126).
Moreover all patients refused categorically total laryngectomy (line 132- and 404-405) and this is a retrospective cohort study.
For this reason we believed that it was not necessary ethics approval from an institutional board.